# Controlled delivery of phosphate to plants with optimized chemical and physical factors

Imani Madison[1] , Sarra P. Darby[2], Perfecto Ascencio[2], Maimouna Abderamane Tahir[3], Lisa Van den Broeck[1], Linh Phan[1], Madison Hooker[1], Timothy Horn[3], Jiangfeng Xu[4], Kirill Efimenko[4], Jan Genzer[4], Juan Claudio Nino[2] and Rosangela Sozzani[1]

[1]Plant and Microbial Biology Department and NC Plant Sciences Initiative, North Carolina State University, Raleigh, NC, USA; [2]Materials Science and Engineering Department, University of Florida, Gainesville, FL, USA; [3]Mechanical and Aerospace Engineering Department, North Carolina State University, Raleigh, NC, USA; [4]Department of Chemical and Biomolecular Engineering, North Carolina State University, Raleigh, NC, USA

## Original Research Article

**Keywords:**
3D bioprinting; bioprinting optimization; phosphate starvation; phosphate fertilizer.

**Corresponding author:**
Rosangela Sozzani;
Email: ross_sozzani@ncsu.edu

**Associate Editor:**
Dr. Ali Ferjani

## Abstract

Sustainable phosphorus fertilization is a growing challenge in agriculture. Phosphorus is necessary for plant growth, but it is typically only bioavailable in its orthophosphate form. Phosphate fertilizers contribute to environmental damage as they leach into aquatic ecosystems. Therefore, it is imperative to develop new fertilization techniques such as controlled-release small-scale phosphate fertilizers. However, iteratively optimizing various new fertilizers using a comparable method is difficult. Here, we use three-dimensional bioprinting as a high-throughput screening platform to evaluate cellular phosphate uptake of various phosphate sources, including triple super phosphate, diammonium phosphate and struvite, which are composed of different chemistries and scales. As a result, we identified ideal phosphate fertilizer sources for the development of controlled-release phosphate fertilizers. Then, we evaluated whether plant growth and root architecture responded differently to the ideal controlled-release fertilizers. This study demonstrates the utility of this screening platform in developing a controlled-release phosphate fertilizer that effectively provides phosphate to plants at the microparticle scale.

## 1. Introduction

Sustainable nutrient management is a growing challenge in agriculture, especially with respect to phosphorus (Bacheva et al., 2024; do Nascimento et al., 2018). Phosphorus is a necessary macronutrient for many metabolic processes needed for plant growth and development (Dong et al., 2017; Gonçalves et al., 2020; Lizcano-Toledo et al., 2021). Phosphorus homeostasis in plants is also linked with plant maintenance of carbon and nitrogen balances (Hermans et al., 2006; Lizcano-Toledo et al., 2021; Syers et al., 2008). Phosphorus starvation in plants promotes extensive local adaptations to the root architecture to optimize phosphorus uptake from the rhizosphere (Al-Ghazi et al., 2003; Hermans et al., 2006; Madison et al., 2023; Péret et al., 2011). These adaptations include inhibited primary root growth and promoted lateral root growth (Péret et al., 2011). Lateral root formation, or patterning, is primarily determined by plant age, but also exhibits developmental plasticity in response to environmental conditions such as phosphate starvation (van Norman et al., 2013). Consequently, Pi-starved roots form a highly branched lateral root system (Hermans et al., 2006). Phosphorus is primarily taken up in orthophosphate (Pi) species, but most phosphorus in soils, or legacy phosphorus, is either present in other biounavailable compounds or is complexed to other mineral cations, such as iron, calcium, aluminium or organic compounds, making soil phosphorus largely inaccessible to plants (Bindraban et al., 2020; Lizcano-Toledo et al., 2021; Madison et al., 2023; Zhu et al., 2018). Moreover, soil phosphate concentration can be as low as 0.65 nM and up to 10 μM, but plant roots need at least 1 μM free orthophosphate in the rhizosphere (Lizcano-Toledo et al., 2021; Péret et al., 2011; Syers et al., 2008). Therefore, fertilizers containing phosphate are a necessary component of agriculture. However, phosphate from fertilizers, sewage and other sources has

largely leached into aquatic ecosystems through runoff, rather than being absorbed by plants, contributing to algal blooms that create hypoxic conditions that kill aqueous wildlife (Dodds & Smith, 2016; Lizcano-Toledo et al., 2021; Schindler, 1974; Syers et al., 2008; van de Wiel et al., 2016). Some studies have found that up to 80% of applied phosphorus has been lost due to leaching, largely due to precipitation (Avila-Quezada et al., 2022; Hertzberger et al., 2021). As we advance, innovations in phosphate fertilizer techniques will facilitate sufficient phosphorus delivery to crops with minimized environmental impact (Bindraban et al., 2020).

Commonly used phosphate fertilizer sources vary in chemical composition and scale, which may influence soil phosphate uptake (do Nascimento et al., 2018; Hertzberger et al., 2021). Superphosphates and ammonium phosphate fertilizer sources are among the most soil-soluble sources that have been implicated as large contributors to phosphorus leaching into aquatic systems (Basavegowda & Baek, 2021; Lizcano-Toledo et al., 2021). Diammonium phosphate (DAP, $(NH_4)_2HPO_4$) contains phosphate and ammonia, while triple super phosphate (TSP, $Ca(H_2PO_4)_2 \cdot H_2O$) contains calcium phosphate (Lizcano-Toledo et al., 2021). Both TSP and DAP are highly soluble in soils, which contributes to their high degree of leaching. Moreover, DAP application to agricultural systems contributes to the production of nitrous oxide, a greenhouse gas, potentially contributing to additional environmental harm (Lizcano-Toledo et al., 2021). Innovations could be made to limit pollution by optimizing the controlled release of superphosphates or ammonium phosphates. Moreover, controlled-release phosphate delivery systems that contain phosphate on the microscale or nanoscale could even reduce the amount of applied phosphates to limit phosphate buildup in soils. Struvite ($MgNH_4PO_4 \cdot 6H_2O$) contains magnesium, ammonium and phosphate and is less soluble than DAP or TSP, resulting in potentially lower levels of leaching (Lizcano-Toledo et al., 2021). Therefore, struvite could be an alternative phosphate source for controlled-release systems, especially since it is recoverable from wastewater treatment plants (Hertzberger et al., 2021). Factors such as scale may also influence phosphate solubility. For example, smaller granule sizes may enhance phosphorus uptake from struvite due to their higher specific areas (Hertzberger et al., 2021).

Recently, fertilizer innovations and alternative fertilization methods have been explored, ranging from nanofertilizers to controlled-release fertilizers (Avila-Quezada et al., 2022; Basavegowda & Baek, 2021; Syers et al., 2008; Wu & Li, 2021; Yadav et al., 2023). These techniques involve encapsulating nutrients in an organic or inorganic material to either prevent immediate release of the nutrient or to deliver nutrients only to target plant tissues (Basavegowda & Baek, 2021; Yadav et al., 2023). Ideally, controlled-release fertilizers will encapsulate and then slowly deliver phosphate more efficiently to plant tissues while reducing the phosphorus levels that leach into the environment (Zulfiqar et al., 2019). Phosphate fertilizers are modified down to a small scale and controlled-release properties, which involves either top-down, chemical or even biological approaches (Zulfiqar et al., 2019). Top-down approaches employ grinding or milling of phosphate sources into smaller particles, but bottom-up approaches employ chemical synthesis of phosphate fertilizer particles to the desired scale and chemical formulation (Zulfiqar et al., 2019). Top-down approaches are rapid, with more variable results. However, bottom-up approaches result in more consistent particle sizes but are more resource-intensive to synthesize (Zulfiqar et al., 2019). In current nanofertilizer development applications, synthesized

bottom-up approaches have generally been more effective than top-down approaches (Zulfiqar et al., 2019). Since phosphate fertilizer particles can be developed in a myriad of ways and scales, it will be useful to evaluate their effectiveness to identify the best options for developing a controlled-release fertilizer (Zulfiqar et al., 2019). Controlled-release fertilizers are synthesized by coating the fertilizer in a material that must degrade before the fertilizer is released (Lawrencia et al., 2021). Coatings such as alginate and starch can retain water, preventing early phosphate release (Lawrencia et al., 2021).

As the development of alternative fertilizer sources or techniques increases, it will be necessary to have more standardized, iterative, high-throughput methods of optimizing controlled-release phosphate fertilizers for controlled phosphate release (Lawrencia et al., 2021; Yadav et al., 2023). Three-dimensional (3D) bioprinting is an emerging technology that can now be used to help develop new phosphate fertilizer sources. Specifically, extrusion-based 3D bioprinting is an additive manufacturing technique that extrudes protoplasts embedded in a customizable bioink to form 3D constructs, forming a structure analogous to a 3D cell culture (Jose et al., 2016; Madison et al., 2025; Mahmood et al., 2022; Ngo et al., 2018; Van den Broeck et al., 2022). Bioink typically includes liquid media and a scaffold that solidifies after bioprinting, keeping the cells in fixed positions within the construct. Over time, the cellular microenvironment can be manipulated as cells grow and divide, facilitating studies of cellular responses to phosphate starvation and different phosphate particles (Figure 1). Moreover, any detrimental effects on plant cells that may be masked in whole plant tests can be detected and managed early. Some potential limitations are inherent to 3D bioprinting, such as excessive cell death caused by suboptimal printing parameters and physical shear stresses induced by extruding cells through a needle (López-Marcial et al., 2018). We have previously optimized cell viability and microcallus formation in a 3D bioprinting pipeline using *Arabidopsis* root, shoot and soybean embryo protoplasts (Van den Broeck et al., 2022). We expand on this by identifying additional factors influencing bioprinted cell viability. Since these parameters are customizable and programmable, these limitations can be overcome by testing and identifying optimal parameter combinations (Madison et al., 2025; Van den Broeck et al., 2022). For example, many possible scaffolding agents exist, such as Pluronic, alginate or agarose, but they are not necessarily equally effective in all given applications. Pluronic, for example, has been extensively used in 3D bioprinting but is not optimally stable within aqueous solutions (López-Marcial et al., 2018; Müller et al., 2015). On the other hand, agarose and alginate are seaweed-derived scaffolds that are not biodegradable or bioactive (Bao et al., 2018; Guimarães et al., 2022; Jovic et al., 2019). Alginate solidifies by crosslinking with cations, such as calcium, while low-melting agarose self-gelates as it cools below $37°C$ (Bao et al., 2018; Jovic et al., 2019). However, Pluronic can achieve linear lines or patterns in prints, whereas agarose or alginate-based hydrogels typically spread beyond the initial print area (López-Marcial et al., 2018).

Here, we demonstrate how 3D bioprinting can be optimized for plant cell assays and then used to screen various phosphate fertilizer sources and to test a new controlled-release phosphate system. First, we identified an optimal scaffold and set of optimal bioprinting parameters for quantifying cell viability, such as extrusion pressure and needle gauge, with Tobacco Bright Yellow 2 (BY-2) cells due to the benefits of their homogeneity and large biomass for many parameter optimization applications (Calcutt et al., 2021; Nagata et al., 1992; Santos et al., 2016). Next, we

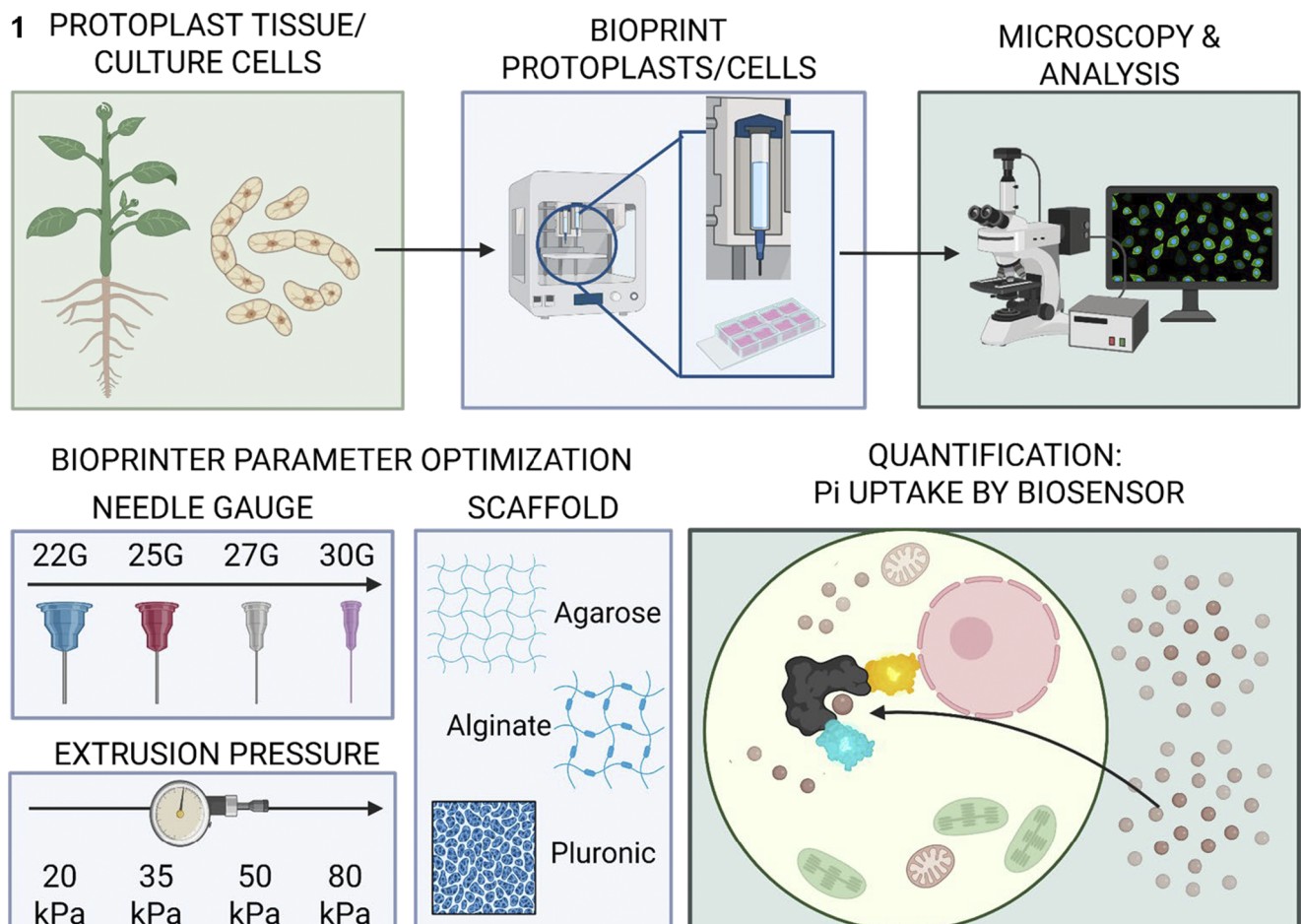

**Figure 1.** Schematic representation of the 3D bioprinting workflow.
Cells from the Tobacco BY-2 cell culture line or root protoplasts from *Arabidopsis* seedlings were bioprinted in eight-well chamber slides, according to each tested parameter or condition, and then imaged and analysed to quantify cellular metrics. The tested parameters included extrusion pressure (20, 30, 50 or 80 kPa), bioink hydrogel composition (low-melting agarose, alginate and Pluronic) and needle gauge (22G, 25G, 27G or 30G). Moreover, the feasibility of using 3D bioprinting to detect cellular responses to changes in the microenvironment at an optimal set of parameters was quantified using phosphate-starved (−Pi) or phosphate-sufficient (+Pi) bioink. Quantification of phosphate uptake was then evaluated using bioprinted cells isolated from a Pi-biosensor line containing a phosphate-binding protein bound at either terminus to an eCFP or VENUS protein that exchanges FRET energy more efficiently if phosphate does not bind the sensor. Conversely, energy is not transferred when phosphate is bound to the protein, as depicted.

identified phosphate starvation root growth responses in *Arabidopsis thaliana* seedlings and cell viability and division responses of 3D bioprinted *Arabidopsis* root protoplasts using the optimal bioprinting parameters. Then, we screened bioprinted *Arabidopsis* protoplasts for differential phosphate uptake of TSP, DAP and struvite that were either in their typically used forms or in their forms ground to the microparticle scale using either a unitary ball mill or a mortar and pestle (M&P). Based on these results, we then developed and screened a controlled-release phosphate fertilizer to ensure that phosphate release was not premature. Finally, we evaluated seedling growth, particularly root architecture, to identify how phosphate chemistry influences root development and local phosphate starvation responses.

## 2. Results

### 2.1. Identifying optimal 3D bioprinting parameters for target applications

To evaluate if scaffold and needle gauge parameters influence cell viability, we bioprinted Tobacco BY-2 cells with each scaffold:

low-melting agarose, alginate and Pluronic, using a 30G needle at 20 kPa and quantified cell viability that same day (Figure 2a). Cell viability in response to agarose was significantly lower than when Pluronic and alginate were used. We then bioprinted BY-2 cells with each scaffold using a 27G needle at 20 kPa and quantified cell viability 2 days later (Figure 2b). Cell viability in response to agar was higher than in response to Pluronic and alginate. This revealed that, while scaffolding has some impact on cell viability, needle gauge may have a stronger influence since the trends in cell viability were reversed when a different needle gauge was chosen, which was not likely caused by a 2-day difference in time. Across both cases, agarose resulted in the highest average cell viability. We then quantified how needle gauge and pressure influence BY-2 cell viability in agarose. BY-2 cells were bioprinted with the needles 22G, 25G, 27G and 30G, and pressures 20, 35, 50 and 80 kPa (Figure 2c). At 20 kPa, cell viability was highest with a 25G needle. At 35 kPa, cell viability was highest with either a 27G or 30G needle. At pressures higher than 35 kPa, discrete constructs did not form well in all needles, except for the 30G needle. At 80 kPa, cell viability was much lower than at 50 kPa using the 30G needle, likely due to the large amount of shear stress applied by pressure

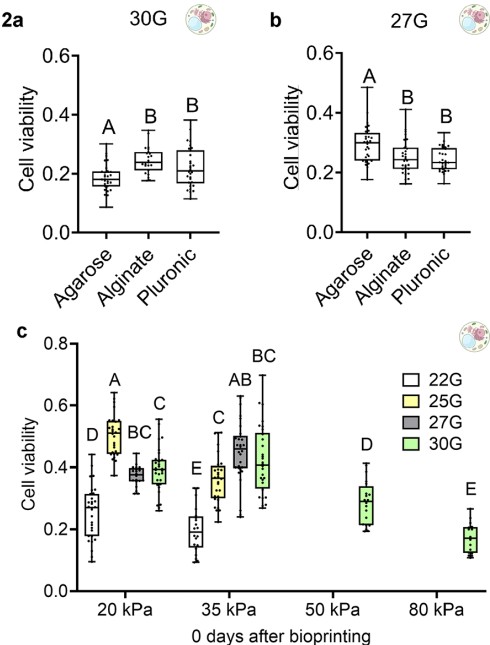

**Figure 2.** Cell viability of 3D bioprinted BY-2 cells in various scaffolds, needle gauges and extrusion pressures.
(a) Cell viability of agar, alginate and Pluronic bioprinted with a 30G needle and imaged afterwards on the same day (D0) in three biological replicates. Statistical analysis was performed using an ANOVA/Tukey's HSD test. (b) Cell viability of agar, alginate and Pluronic bioprinted with a 27G needle and imaged 2 days afterwards in three biological replicates. Statistical analysis was performed using an ANOVA/Tukey's HSD test. (c) Cell viability of BY-2 bioprinted cells using each parameter combination: 22G, 20 kPa (biological replicates $n = 3$); 22G, 35 kPa ($n = 3$); 25G, 20 kPa ($n = 3$); 25G, 25 kPa ($n = 3$); 27G, 20 kPa ($n = 3$); 27G, 35 kPa ($n = 3$); 30G, 20 kPa ($n = 3$); 30G, 35 kPa ($n = 3$); 30G, 50 kPa ($n = 2$); and, 30G, 80 kPa ($n = 2$). Statistical analysis was performed using an ANOVA/Tukey's HSD test.

this high. Across all factors, cell viability was highest using a 25G needle at 20 kPa. However, the cell viabilities resulting from the use of 27G and 30G needles were consistently similar across both 20 and 35 kPa, whereas cell viability dropped between 20 and 35 kPa in response to the 22G and 25G needles. These data suggest that optimal needle gauges are different at different pressures, so needle selection should depend on the pressure required to extrude the bioink and cells of interest.

### 2.2. Quantifying phosphate starvation responses at the cellular level

Since phosphate starvation impacts seedling root growth and bioprinted cell growth and division, we evaluated whether 3D bioprinting could capture these known cellular phosphate starvation responses in *Arabidopsis* (Al-Ghazi et al., 2003; Lai et al., 2007; Madison et al., 2023; Tian et al., 2021). We bioprinted *Arabidopsis* root protoplasts using a 30G needle at 20 kPa with agarose. Then, cell viability at days 0–3 (Figure 3a) and at days 7 and 10 (Figure 3b) after bioprinting was quantified. Cell viability remained statistically similar across 0–3 days using either +Pi or −Pi bioink, suggesting that bioprinting did not impair cell viability between each condition. However, cell viability dropped significantly at both 7 (7d) and 10 days (10d) using −Pi bioink. Moreover, the pCYCB1::CYCB1-GFP marker line (Buckner et al., 2019; Van den Broeck et al., 2022) was bioprinted to quantify the percentage of single, dividing or multicellular meristematic

cells actively entering mitosis at 7d and 10d after bioprinting (Figure 3c). In response to −Pi bioink, there was a significantly greater percentage of single cells than multicellular complexes that were actively dividing at both 7d and 10d, suggesting that cell division had been delayed by phosphate starvation.

To compare cellular phosphate responses to plant phosphate growth and cell division responses, we grew *Arabidopsis* seedlings on phosphate-starved (0 mM Pi, −Pi) or phosphate-sufficient (standard MS, +Pi) media and then evaluated root growth from 3 to 14 days afterwards (Figure 4d). As expected, −Pi primary roots grew significantly shorter from day 8 onwards (Al-Ghazi et al., 2003). Then, we quantified aspects of meristematic cell division in roots, such as the length of the meristematic zone (Figure 3e) and the number of cortex cells within the meristematic zone in 7d and 10d roots (Figure 3f). In both 7d and 10d −Pi roots, the meristematic zone and number of cortex cells were reduced. Moreover, the 10d +Pi roots had more cortex cells and a longer meristematic zone than at 7d, whereas neither metric did not changed in −Pi roots. This suggests that Pi-starvation results in reduced meristematic cell division (Lai et al., 2007; Péret et al., 2011). These data also suggest that 3D bioprinted cells retain a similar pattern of reduced cell division in −Pi bioink as observed in whole root meristematic cells, and that they are actively responding to −Pi bioink.

### 2.3. High-throughput screening of phosphate fertilizer sources of differing chemistries and scales

Since phosphate starvation and phosphate sufficiency could be distinguished using 3D bioprinting of plant cells, we then screened phosphate fertilizer source particles for differential cellular phosphate uptake using phosphate particles suspended in −Pi bioink. We acquired an *Arabidopsis* Pi-biosensor that reports a low Förster Resonance Energy Transfer (FRET) binding efficiency or FRET ratio when phosphate binds its phosphate-binding protein that has a donor and acceptor fluorophore at either terminus (Figure 1) (Mukherjee et al., 2015). This biosensor has been used in past studies to quantify cellular phosphate levels in different root developmental zones and after Pi resupply to plants (Mukherjee et al., 2015; Sahu et al., 2020). To determine the sensitivity of the sensor to different phosphate concentrations, we performed FRET imaging on 7d old Pi-biosensor seedlings grown on the following phosphate concentrations: 0, 0.25, 0.5, 0.75 and 1.2 mM Pi (Figure 4a). Between 0 and 0.5 mM Pi and between 0.5 and 1.2 mM Pi, the Pi-biosensor was significantly high and low, respectively, suggesting that the Pi-biosensor sensitivity to −Pi conditions is most distinct below 0.5 mM Pi. Then, we selected the following phosphate fertilizer sources: TSP, DAP and struvite, to evaluate whether intracellular phosphate concentrations vary based on phosphate fertilizer chemistry. We also prepared phosphate fertilizer particles as either M&P-ground, unitary ball-milled (UBM), synthesized(Synth.), or as received (as-is) in their commercially available forms, to evaluate whether particle uniformity or size at the microscale influences intracellular phosphate concentration differentially as well. These particles were incorporated into the bioink scaffold at 10 mg/mL to provide more than minimally sufficient phosphate levels. The Pi-biosensor seedlings were grown on 0 mM Pi media before protoplasting. Then, protoplasts were isolated from the seedlings at 7d after sowing and bioprinted with −Pi bioink suspended with either phosphate fertilizer particle. Then, FRET binding efficiencies were determined 3 days after bioprinting (Figure 4b). Pi-biosensor protoplasts were also bioprinted with only +Pi bioink and −Pi bioink as controls. To establish a baseline of FRET efficiencies corresponding

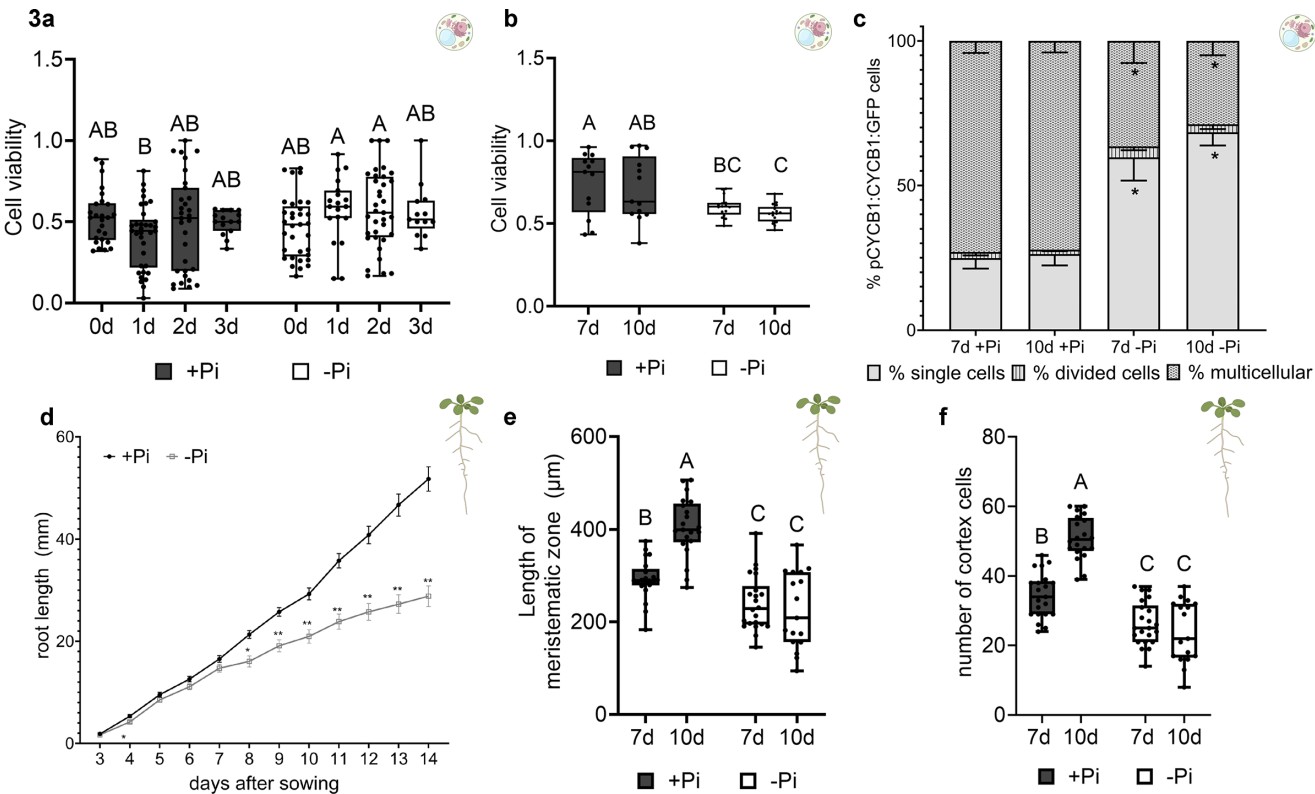

**Figure 3.** Root growth and bioprinted cell division responses to phosphate starvation.
(a) Cell viability of Col-0 root cells between 0 and 3 days after bioprinting in either +Pi or −Pi bioink. Statistical analysis was performed using the ANOVA/Tukey's HSD method across three biological replicates, and statistically significant groups are indicated by different letters. (b) Cell viability of Col-0 root cells at 7 and 10 days after bioprinting in either +Pi or −Pi bioink. Statistical analysis was performed using the ANOVA/Tukey's HSD method across three biological replicates. (c) Percent of cell division in pCYCB1:CYCB1:GFP bioprinted cells at 7 and 10 days after bioprinting in either +Pi or −Pi bioink. Cells were scored as either single cells, divided cells (consisting of a mother and daughter cell) or microcallus, and then expressed as a percentage of the total cells present. ∗ denotes significant differences of −Pi values from +Pi values based on Steel–Dwass pairwise comparisons on each day between +Pi and −Pi across three biological replicates. Pairwise comparisons were also performed between 7 and 10 days at either the +Pi or −Pi condition, and no significant differences were observed. (d) Root length of Col-0 roots grown in either Pi-sufficient (+Pi) media or Pi-starvation (−Pi) media and root length was measured each day after day 3 of sowing until 14 days after sowing. Statistical analysis is a pairwise Student's *T*-test comparing root length between +Pi and −Pi at each time point and includes at least three biological replicates (∗$p < 0.01$ and ∗∗$p < 0.0001$). (e) Length of the root meristematic zone after 7 and 10 days of growth on either Pi-starvation or Pi-sufficient media. The length of the meristematic zone was measured at both 7 and 10 days after sowing. Statistical analysis is an ANOVA/Tukey's HSD analysis across three biological replicates. (f) Number of cortex cells within the meristematic zone after 7 and 10 days of growth on either Pi-starvation or Pi-sufficient media. Statistical analysis is an ANOVA/Tukey's HSD analysis across three biological replicates.

to optimal cellular Pi concentrations, seedlings were grown in +Pi media and bioprinted in +Pi bioink. As expected, the −Pi bioink resulted in the highest FRET efficiencies, while the +Pi bioink and baseline were both low and similar to each other. All of the TSP particles, except for synthetic TSP, had FRET efficiencies that were not low enough to suggest optimal baseline FRET efficiencies. The as-is DAP and synthetic DAP FRET efficiencies were also too high to suggest optimal intracellular Pi. However, ball-milled and M&P-ground DAP FRET efficiencies were similar to the baseline. Finally, the Synth. struvite FRET efficiency was also similar to the baseline. These data suggest that DAP and Synth. struvite are generally optimally taken up by cells on the microscale level and would be good candidates for developing a controlled-release phosphate fertilizer.

### 2.4. Quantification of cellular Pi uptake and seedling root architecture from coated controlled-release phosphate particles

We coated each DAP and struvite particle type with a 12% alginate-corn starch mixture to form controlled-release phosphate beads. We then bioprinted the Pi-biosensor and added 1 bead to each construct to quantify the resulting FRET efficiencies from day 0 to

3 after bioprinting (Figure 4c). As expected, the −Pi bioink resulted in high FRET efficiencies by day 1 onwards. FRET efficiencies corresponding to coated as-is DAP and M&P-ground DAP peaked on days 1 and 2, respectively, before dropping to baseline levels. This suggests that the phosphate contained in the bead had been released earlier than desired. FRET efficiencies corresponding to coated ball-milled and Synth. DAP and Synth. struvite all increased steadily across the time course, suggesting that much of the phosphate had remained within the beads. However, coated synthetic DAP resulted in inconsistent increases and decreases in FRET efficiencies over time. So, coated ball-milled DAP and synthetic struvite resulted in the ideal trends over time in which phosphate was not released from the beads to cells at early time points.

We evaluated whole plant root growth in response to coated DAP and coated struvite. Thirty-two coated DAP or coated struvite beads were placed on −Pi media in the same region as the sown seeds. Then, the resulting root architecture characteristics, such as primary root growth, lateral root number, average lateral root length and lateral root branching density, were measured after 14 days (Figure 4d–g). At days 7, 10 and 14 after sowing, +Pi primary roots were longer than −Pi primary roots (Figure 4d). By day 7, primary root length with coated DAP or struvite, however,

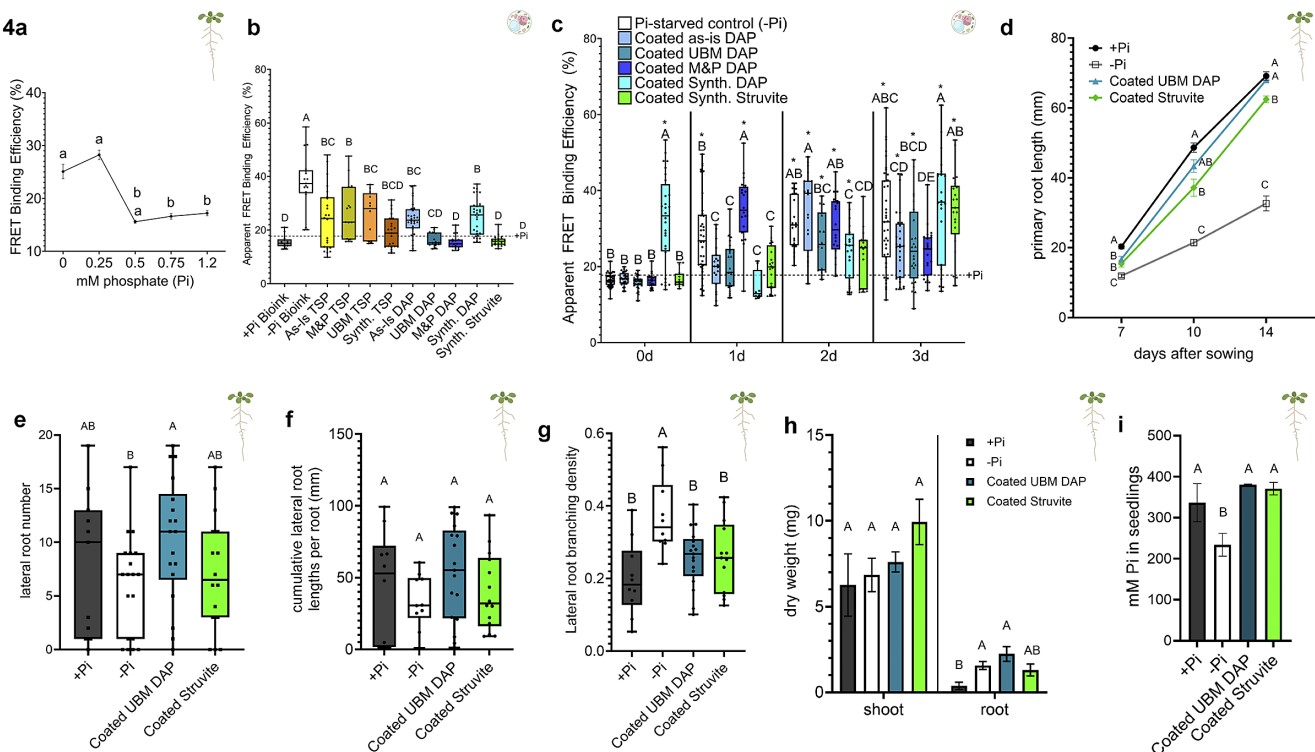

**Figure 4.** Quantification of phosphate uptake from coated Pi by bioprinted biosensor cells and the resulting root architecture in seedlings.
(a) FRET binding efficiency in cells of whole roots grown on either 0, 0.25, 0.5, 0.75 or 1.2 mM Pi. Statistical analysis is pairwise compared to either 0 or 1.2 mM Pi. a = statistically different from 1.2 mM Pi; b = statistically different from 0 mM Pi. (b) FRET binding efficiency in bioprinted phosphate biosensor cells in three biological reps. Statistical analysis was performed using an ANOVA/Tukey's HSD test across all groups. Biosensor cells were bioprinted with no phosphate (−Pi), standard cell culture concentrations of potassium phosphate (+Pi), triple super phosphate (TSP) and diammonium phosphate (DAP) particles at varying formulations: ground (M&P), milled (UBM) or synthesized (Synth.), as well as synthesized struvite. The Pi-sufficient baseline (dotted line) was determined by FRET imaging of bioprinted biosensor cells isolated from plants grown on standard MS media for 7 days and bioprinted with +Pi bioink (n = 49). Unique letters indicate significant differences. (c) Apparent FRET efficiency of coated DAP and struvite particles prepared either as received (as-is) or using a mortar and pestle, or a unitary ball mill, at days 0–3 after bioprinting. The Pi-sufficient baseline (dotted line) was determined by FRET imaging of bioprinted biosensor cells isolated from plants grown on standard MS media for 7 days and bioprinted with +Pi bioink. Unique letters indicate significant differences between samples at each time point. Asterisks indicate significant differences from baselines (p < 0.001). (d) Primary root growth of seedlings grown in +Pi, −Pi, coated DAP and coated struvite-containing media. Roots were measured from 3 days until day 7, 10 and 14 after sowing. (e) Primary root growth of seedlings grown in +Pi, −Pi, coated DAP and coated struvite-containing media. Lateral roots were counted at day 14 after sowing. (f) Cumulative lateral root length of seedlings grown in +Pi, −Pi, coated DAP and coated struvite-containing media. Lateral roots were measured at day 14 after sowing (g).
Lateral root branching density of seedlings grown in +Pi, −Pi, coated DAP and coated struvite-containing media at day 14 after sowing. (h) Shoot or root dry weight of seedlings grown in +Pi, −Pi, coated UBM DAP or coated struvite media for 14 days in three biological replicates. Statistical analysis is a Student's T-test, and unique letters denote statistical significance. (i) Phosphate concentration in seedlings grown in +Pi, −Pi, coated UBM DAP or coated struvite media for 14 days in three biological replicates. Statistical analysis is a Student's T-test, and unique letters denote statistical significance.

was of intermediate length between that of +Pi and −Pi. However, by days 10 and 14, the primary root length of seedlings grown with coated DAP beads was similar to that of +Pi, while the primary root length with coated struvite was still shorter than that of +Pi but longer than that of −Pi. This suggests that primary root growth in seedlings grown with coated DAP is similar to Pi-sufficient roots and does not exhibit a Pi starvation phenotype. Moreover, the lateral root number of +Pi and −Pi seedlings was similar to each other, but the lateral root number in response to coated DAP was larger than in roots grown in −Pi conditions (Figure 4e). The average lateral root length per root was also similar across all samples (Figure 4f). However, the lateral root branching density was high in −Pi roots and low in +Pi, coated DAP and coated struvite roots (Dubrovsky & Forde, 2012) (Figure 4g). This suggests that the distribution of lateral roots in roots grown with both coated DAP and coated struvite beads was similar to that of +Pi roots, so this aspect of root architecture was not perturbed by application of either coated DAP or struvite particles. Overall, these data suggest that the application of coated DAP ultimately resulted in a root architecture most consistent with Pi-sufficiency,

while seedlings treated with coated struvite had a mixture of root architecture characteristics indicative of intermediate root adaptations to phosphate starvation. We then quantified the plant biomass and phosphate concentration in response to coated UBM DAP or struvite particles. Shoot dry weights of 14-day-old seedlings grown in +Pi, −Pi, coated UBM DAP and coated struvite media were all similar (Figure 4h). As expected, root dry weight of seedlings grown in +Pi media was significantly lower than that of seedlings grown in −Pi media, suggesting more biomass allocation to roots under Pi starvation (Aziz et al., 2014). Moreover, the root dry weight of seedlings grown with coated UBM DAP was similar to that of −Pi seedlings, but the root dry weight of seedlings grown with coated struvite was intermediate, similar to that of +Pi or −Pi seedlings. This suggests that the biomass distribution of seedlings grown with coated struvite was most similar to +Pi seedlings, while seedlings grown with coated DAP allocated more biomass to root tissues than +Pi seedlings did. Moreover, seedlings grown in coated UBM DAP had the highest average lateral root numbers (Figure 4e), which may have contributed to the higher root dry weight of seedlings grown in coated UBM DAP (Figure 4h). Finally, we performed an

inorganic phosphate quantification assay in whole seedlings grown in +Pi or −Pi media or with coated UBM DAP or coated struvite beads. As expected, −Pi plants had the lowest concentrations of phosphate (Figure 4i). However, seedlings grown with coated UBM DAP or coated struvite beads had similar phosphate concentrations to +Pi seedlings. These data suggest that seedlings were able to take up sufficient amounts of phosphate released by either coated phosphate particle type, with few stress-induced local adaptation responses by the roots.

## 3. Discussion

Nutrient uptake is a complex process that relies on various factors, such as the chemical rhizosphere environment, nutrient solubility and scales (Hertzberger et al., 2021). Here, it has been shown that phosphate particle size and chemistry can influence cellular phosphate uptake. For example, in this study, Synth. struvite, ground DAP, ball-milled DAP and Synth. TSP were taken up by individual cells at Pi-sufficient levels within 3 days after bioprinting. However, the remaining sources were not taken up at Pi-sufficient levels. Treatment with coated DAP and struvite particles in whole plants resulted in similar root growth phenotypes to Pi-sufficient plants. Specifically, coated DAP particles resulted in primary and lateral root development most consistent with Pi sufficiency, while treatment with coated struvite particles resulted in Pi-sufficient lateral root architecture and slightly shorter root growth that was still longer than Pi-starved roots. When evaluating different phosphate fertilizer sources, a screening system like 3D bioprinting can help identify any differences on the cellular scale that may be masked at the plant scale, and will help identify ideal sources comparably.

In future studies, even more chemical factors can be evaluated, such as the presence of these phosphate fertilizer sources in the presence of other oxides and cations, like iron, aluminium and sodium, that could negatively influence cellular phosphate uptake. For example, iron oxides and aluminium oxides in soil bind phosphate and reduce its bioavailability to cells (Avila-Quezada et al., 2022; do Nascimento et al., 2018). Moreover, different chemical fertilizer formulations may induce different regulatory mechanisms (Tian et al., 2021). For example, ammonium and phosphate interact to promote root elongation, and ammonium is also involved in upregulating phosphate starvation response genes and rhizosphere acidification proteins to solubilize soil nutrients (Tian et al., 2021). Moreover, calcium phosphate can decrease the upregulation of phosphate uptake mechanisms when in the presence of ammonium (Tian et al., 2021). Therefore, future work will identify which regulatory mechanisms and crosstalk are induced by different fertilizer chemistries.

One potential benefit to incorporating soluble phosphate fertilizer sources like ammonium phosphates or low-soluble phosphate fertilizer sources like struvite into granular controlled-release beads is to reduce their solubility and leaching potential while keeping the granular form that is more compatible with application implements in the field than are powdered forms (Hertzberger et al., 2021). These data also provide direction into how struvite can be leveraged as an alternative phosphate fertilizer source, especially when developing controlled-release particle fertilization systems. Other potential avenues for the development of controlled-release phosphate systems include creating blends with struvite and more soluble ammonium phosphate fertilizers to improve solubility for both late and early phosphorus delivery, if desired (Hertzberger et al., 2021).

This investigation demonstrates that a standardized screening system, such as 3D bioprinting, can be useful in evaluating a wide range of phosphate sources and scales in plant cells in the future to help identify the advantages or disadvantages of each source. The parameter optimization performed in this study clarifies several possible bioprinting parameter choices in downstream applications. If high pressures are needed, a 30G needle should be used with the expectation that cell viability would be low. If cells of similar sizes to BY-2 cells are needed, then a 25G needle at 20 kPa would be optimal. However, if pressure higher than 20 kPa is needed, then either a 27G or 30G needle would be optimal. Or, if the desired pressure is unknown, then a 27G or 30G needle should be used at pressures between 20 and 35 kPa. For smaller cells like in *Arabidopsis* protoplasts, the shear stresses from either gauge needle would be less intense, so the effect on cell viability would be less drastic. Therefore, this data represents an upper limit on needle and pressure combinations. Since 3D bioprinting is customizable and compatible with plant protoplasts, it can be used in future work to evaluate differential phosphate uptake or cellular growth and division in crops and plants outside of *Arabidopsis* to help develop targeted fertilization methods in specific agricultural systems and environmental contexts (Van den Broeck et al., 2022; Bacheva et al., 2024). The customizable nature of 3D bioprinting also means that the uptake of other nutrients, such as nitrogen and potassium, or other biologicals or additives, can be evaluated in isolation or in combination with phosphate to further develop other sustainable methods of fertilization in agriculture. The benefits of this platform also include the customizability of the bioink, which can have any combination of hormones, nutrients and compounds directly mixed with plant cells so that any newly developed controlled-release phosphate fertilizer or phosphate fertilizer source can be directly bioprinted with live plant cells to determine the best options before investing extensive time and resources into whole plant and field trials.

Going forward, the timing of the controlled-release phosphate fertilizer source should be evaluated at longer time scales and in whole plants in future studies to determine how long it takes for phosphate to be completely released. This will be critical since a controlled-release fertilizer should not release the majority of its fertilizer too early after application, but over a longer duration after application (Lawrencia et al., 2021; Sano et al., 1999; Sempeho et al., 2014). Additional studies of phosphate uptake under combinatorial stress, including phosphate starvation with other abiotic stresses, will likely expand the requirements for efficient controlled-release phosphorus fertilization. Crop resilience to abiotic stress, such as water stress, salinity, temperature and heavy metal stress, involves phosphorus homeostasis (Khan et al., 2023). Therefore, the optimal amounts and rate of controlled release of phosphate applied to crops experiencing multiple stresses may be higher than what is minimally necessary for crops only experiencing phosphate starvation. So, optimizing crop resilience to systemic environmental dynamics will be an important step going forward (Ulanowicz, 2003). Regulations are currently being outlined to define thresholds and timelines of fertilizer release from controlled-release fertilizers (Lawrencia et al., 2021). Future research should evaluate the performance of controlled-release fertilizers using relevant guidelines. Future development of new fertilizers should also be tailored to different types of geographical regions, especially in developing countries, due to significant differences in crop species, soil composition, geopolitical constraints and environmental impacts (Chuma et al., 2022; Chowdhury et al., 2017; Bacheva et al., 2024; Syers et al., 2008). While controlled-release fertilizers still have

potential limitations like systemic and long-term studies on safety, environmental impacts and fate in plants and the broader ecosystem, the results from this study present a 3D bioprinting platform with which they can be iteratively optimized to overcome any limitations discovered going forward as more information is obtained (Wu & Li, 2021; Zulfiqar et al., 2019).

## 4. Materials and methods

### 4.1. BY-2 cell culture and isolation

The NT-1 medium for the BY-2 cell culture was made by dissolving 4.3 mL of Murashige and Skoog (MS) Salts, 30 mL of sucrose, 10 mL of B1-Inositol Stock, 3 mL of Miller's I Stock and 0.02 mL of 2,4-Dichlorophenoxyacetic acid (2,4-D) Stock (10 mg/mL in 0.1 M KOH) into 500 mL of distilled water and adjusting the total volume with distilled water up to 1,000 mL. B1-Inositol and Miller's I Stock solutions were made by dissolving 5 g of myo-Inositol and 0.05 g thiamine-hydrochloride up to 500 mL of distilled water, and 30 g of potassium dihydrogen phosphate ($KH_2PO_4$) up to 500 mL with distilled water.

The BY-2 cell suspension culture was isolated from a 60-mL flask of a 7d old *in vitro* BY-2 mother culture, which was replenished weekly with fresh nutrient NT-1 medium and kept under constant light in an orbital shaker at a speed of 150 rpm at 23°C, by transferring 2 mL of cells from the mother culture into a new vial that contains 58 mL of NT-1 medium. The resulting BY-2 cell flask was kept in the same orbital shaker under the same conditions and used for 3D bioprinting on day 3 after isolation.

### 4.2. Plant growth and protoplast isolation

The BY-2 protoplasting solution was made by dissolving 1% (w/v) cellulase (EMD Millipore) and 0.1% (w/v) pectolyase (Sigma-Aldrich) into a 100 mL solution of 99 mL 0.4 M mannitol and 1 mL 500 mM MES. The enzymatic solution was then sterilized using vacuum filter sterilization (0.22 μm). For *Arabidopsis*, the protoplasting solution was similar, except for 1.5% cellulase and 0.1% pectolyase as described in Van den Broeck et al. (2022)). *Arabidopsis* Col-0 seedlings and pCYCB1::CYCB1-GFP (Buckner et al., 2019) *Arabidopsis* seeds were wet sterilized and stratified for 2 days, then grown on MS media for 7d, at which the roots were cut and placed in a 70-μm cell strainer filled with the *Arabidopsis* enzyme solution. After 1 (BY-2) and 2 h (*Arabidopsis*), protoplasts were transferred to a sterile 15 mL conical tube and then centrifuged at 500*g* at 23°C for 5 min. Finally, they were resuspended in Protoplast Induction Media (PIM) as described further in the *Methods* (Van den Broeck et al., 2022). For root growth assays, roots were marked at days 3, 7, 10 and 14 after sowing, then root lengths from day 3 to each time point were measured using ImageJ (Schneider et al., 2012). Lateral root branching density was calculated by dividing the number of lateral roots by the length of the root branching zone, which is from the base of the hypocotyl to the emerged lateral root closest to the root tip. To measure dry weights, plants were dried in an oven at 65°C for at least 6 h until their weight stabilized. Then, plant inorganic phosphate quantification was performed using a molybdate blue assay as described in Land et al. (2021)), using a Varioskan LUX Multimode Microplate Reader at 660 nm.

For BY-2 experiments, 10 mL of the 3d-old BY-2 cell vial is placed into a sterile 50 mL conical tube under the sterile hood, and the tube is then centrifuged at a speed of 500*g* and a temperature of 23°C for 5 min. Then, the supernatant was removed, and 7 mL

of the enzymatic solution was added to the cell pellet. The tube was then transferred to an orbital shaker to rotate for an hour at 85 rpm and 23°C. Once the cell wall degradation had completed, the tube was centrifuged at 500*g* and 23°C for 3 min, and the supernatant was then removed without disturbing the protoplast pellet. Then, the pellet was resuspended in 5 mL NT-1 medium and centrifuged again at 500*g* and 23°C for 3 min. The supernatant was then removed. For the low-melting agarose and alginate scaffold tests, the protoplasts were resuspended in 4 mL of NT-1 medium. For the Pluronic scaffold test, 4 mL of 20% Pluronic was immediately added to the protoplast pellet for bioprinting.

### 4.3. PIM and scaffold composition

Protoplasts were suspended with NT-1 Buffer or PIM before bioprinting. All *Arabidopsis* protoplasts bioprinted in +Pi bioink were formulated. PIM was prepared as described in Van den Broeck et al. (2022)) with ½ B5 media, 1.03% sucrose, 001% MES salts, 2,4-D, BAP (6-benzylaminopurine), calcium chloride hydrate, sodium succinate, folic acid, phytosulfokine and timentin. All *Arabidopsis* protoplasts that were bioprinted with −Pi bioink were resuspended in −Pi PIM formulated with a homemade B-5 salt recipe in which phosphate was not added.

Low-melting (or low-gelling) agarose and alginate scaffolds were kept at 37°C before bioprinting. Pluronic was kept at room temperature for at least 7 min before use, to avoid its solidification. Agarose scaffold was composed of 2.4% agarose in PIM or NT-1. The alginate scaffold was composed of 75% alginate in PIM or NT-1. Pluronic was composed of 20% Pluronic in PIM or NT-1.

### 4.4. Bioprinting and construct maintenance

The corresponding bioinks are prepared with one volume of scaffold material and three volumes of the protoplast resuspended in NT-1 medium. After the addition of the scaffold materials, their bioinks are immediately transferred to their corresponding cartridges and bioprinted to prevent premature solidification of the bioink, which would lead to clogging of the bioink in the bioprinter syringe. Each concentration was bioprinted into an eight-well iBidi chamber slide using a Cellink extrusion 3D Bioprinter. For bioprinting with Pluronic, bioprinting was performed at room temperature. For bioprinting with each needle gauge, the outer diameters were 0.7 mm (22G), 0.5 mm (25G), 0.4 mm (27G) and 0.3 mm (30G). The bioprinter extrusion toolhead was set to a temperature of 37°C for bioprinting with both low-melting agarose and alginate to help maintain the desired viscosity of the bioinks and their printbed at a temperature of 23°C to facilitate gelation of the scaffolds. For all of the experiments involving low-melting agarose and Pluronic, 200 μL of NT-1 was added to each well of the bioprinted slides. Crosslinking was also performed for constructs bioprinted with alginate by incubating 150 μL of calcium dichloride ($CaCl_2$) in each well for 1 min. Then, 200 μL of NT-1 medium was added to the constructs. All the bioprinted samples were kept in the dark in a Percival at 23°C.

### 4.5. Phosphate fertilizer particle milling, synthesis and coating

Commercially available TSP ($Ca(H_2PO_4)_2$, $P_2O_5$ 46%) and DAP 98% (($NH_4)_2HPO_4$, 98%) were acquired from Cz Garden Supply (USA) and Thermo Fisher Scientific Chemicals, Inc. (Ward Hill, Massachusetts), respectively. The chemicals used for the synthesis of TSP, DAP and struvite included phosphoric acid ($H_3PO_4$, 85%, Fisher Scientific), calcium carbonate ($CaCO_3$,

99.5%, Alfa Aesar), ammonium hydroxide (NH$_4$OH, 28–30%, Fisher Scientific), ammonium phosphate monobasic (NH$_4$H$_2$PO$_4$, 99.9%, Fisher Scientific), magnesium acetate aqueous solution (Mg(CH$_3$COO)$_2$, 1 M, Thermo Scientific) and sodium hydroxide solution (NaOH, 50% (w/v), Spectrum Chemical MFG Corp), all used as received without further purification. To produce controlled-release phosphate beads, corn starch ((C$_6$H$_{10}$O$_5$)$_n$ Fisher Scientific), sodium alginate (C$_5$H$_7$O$_4$COONa, Sigma Aldrich) and calcium chloride (CaCl$_2$, ≥97%, Sigma Aldrich) were used.

To prepare the materials, as-is TSP and DAP granules were ground separately using an M&P for 5 min, resulting in an average particle size of ~4.4 and ~20.5 µm, respectively. Subsequently, additional quantities of TSP and DAP were milled separately in a unitary ball mill for 48 h, achieving particle sizes of ~17.3 µm for TSP and ~8.7 µm for DAP.

TSP powders were synthesized by preparing a 50% w/w phosphoric acid (H$_3$PO$_4$) solution from 85.44% phosphoric acid. The solution was heated to 90°C for the water to evaporate, and 4.38 g of 99.99% calcium carbonate was gradually added while stirring at 300 rpm. After stirring the mixture for 1 h, the resulting slurry was dried overnight at 95°C, resulting in an average particle size of ~4.39 µm.

DAP was synthesized by reacting ammonium hydroxide (NH$_4$OH) with phosphoric acid. A 1 M solution of phosphoric acid was prepared from 85.44% phosphoric acid. The exact concentration of the ammonium hydroxide (28–30 wt.%) was determined via titration, and a 1 M solution of ammonium hydroxide was then prepared. The ammonium hydroxide solution was added dropwise to the phosphoric acid solution in a 2:1 molar ratio while stirring at 200 rpm. The mixture was then heated at 80°C to partially evaporate the water and subsequently allowed to cool and crystallize at room temperature for 48 h.

The synthesis of struvite was adapted from Rathod et al. (2015) with modifications (Rathod et al., 2015). Two separate 0.2 M solutions of ammonium phosphate monobasic and magnesium acetate were prepared using deionized water. These solutions were simultaneously added dropwise into a beaker under continuous stirring at 300 rpm at room temperature. After complete mixing, the pH of the solution was adjusted to 10.5 using sodium hydroxide while monitoring with an Accumet Basic AB15 pH meter. Once the pH was adjusted, the reaction mixture was stirred for 1 h to ensure complete precipitation. The precipitate was then collected by filtration using a Whatman 602 H filter paper, washed three times with deionized water and dried at room temperature for 24 h.

The bead preparation procedure was adapted from Phang et al. (2018)). A solution of 12 wt.% corn starch, 1 wt.% sodium alginate and 5 wt.% fertilizer was blended gradually and stirred at 300 rpm for 30 min to ensure a homogeneous mixture. Simultaneously, a 1.5 M solution of calcium chloride was prepared. Using a micropipette calibrated to 1.5 µL, the sodium alginate solution was carefully dispensed, drop by drop, into the continuously stirred 1.5 M CaCl$_2$ solution until all the solution was used. The formed beads were allowed to settle in the solution for 20 min before being removed, rinsed with deionized water and dried at 80°C for 24 h. This same coating process was later applied to all various types of fertilizers, including as-is, M&P, UBM and Synth. TSP, DAP and struvite.

### 4.6. Confocal imaging and image analysis

Microscopic evaluation of BY-2 cells was observed using staining, imaging and digital image analysis. For cell viability assays, staining was done with fluorescein diacetate (FDA) solution, prepared by mixing 5 mg of FDA into 1 mL of Acetone in a 15-mL conical tube, right before each imaging session to stain viable (live) cells. Ten microlitres of 1 M propidium iodide (PI) solution is diluted into 90 µL of sterile distilled water for counter-staining to determine the nonviable (dead) cells. Thus, 10 µL of the diluted PI solution and 2 µL of the FDA solution were added to each samples' wells on each respective imaging time point. After staining, the bioprinted constructs of each sample were imaged using a Zeiss LSM 980 confocal microscope, and 5–10 3D images with varying z-stacks were generated for each sample. The digital analysis of the confocal images was then done as in Van den Broeck et al. (2022)). The image analysis results were the viable and nonviable cell counts in csv files used to create a single long data format csv file, that were further analysed and visualized using a Python script to quantify the experimental results.

Using a Zeiss LSM980 Confocal Microscope, FDA confocal imaging was performed using a 488 nm laser, and collection was detected at 482–570 nm. pCYCB1::CYCB1-GFP imaging was performed using a 488 nm laser, and collection was detected at 482–570 nm. PI imaging was performed using a 561 nm laser, and collection was detected at 570–632 nm. A Leica Stellaris 8 microscope and FRET-FLIM analysis software were used for FRET binding efficiency imaging and analysis. VENUS was excited using a 514 nm laser, and collection was detected at 520–650 nm. Cyan Fluorescent Protein (CFP) was excited using a 448 nm laser, and collection was detected at 445–510 nm. For the +Pi baseline, $n = 49$ cells were imaged. For all bare and coated samples and time points, at least three biological replicates were imaged.

**Open peer review.** To view the open peer review materials for this article, please visit http://doi.org/10.1017/qpb.2025.10023.

**Competing interest.** RS declares a conflict of interest with Raleigh Biosciences.

**Data availability statement.** The *Arabidopsis* lines and particles generated in this study are available upon reasonable request.

**Author contributions.** IM, TH, JCN and RS conceived and designed the study. IM, MAT, LP and MH conducted bioprinting and root growth data gathering. PA and SPD conducted the phosphate milling, synthesis and coating. IM and MAT wrote the article. LVdB, JX, KE, JG, JCN and RS provided critical evaluation during the study and writing.

**Funding statement.** This work was supported by the NSF Science and Technologies for Phosphorus Sustainability (STEPS) grant (CBET-2019435). This work was also supported by the NSF EAGER grant (MCB #2039285) to TJH and RS. Work by IM was supported by the NSF Postdoctoral Fellowship award (IOS-2305774).

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
