## [Reviewer Report]

The research results of this work, submitted by the authors, can be summarized as follows:

A 3D bioprinting platform for plant cells was developed to optimize the impact of parameters such as bioink composition, needle gauge, and extrusion pressure on plant protoplast viability. For example, the use of 27G or 30G needles at pressures of 20-35 kPa was shown to be optimal for maintaining cell viability of Arabidopsis protoplasts.

Phosphate sources with different particle sizes and properties were prepared by processing commercial phosphate fertilizers (TSP, DAP) in a mortar and pestle or unit-type ball mill and further chemical synthesis of TSP, DAP, and struvite.

Coating techniques using cornstarch and alginate were employed to prepare sustained-release beads of phosphate fertilizers and to establish a basis for evaluating phosphate release kinetics at the cellular level.

Bioprinted phosphate biosensor cells, utilizing FRET imaging technology, demonstrated that changes in intracellular phosphate levels, both with and without phosphate, can be visualized and quantified in real-time at the cellular level.

The developed 3D bioprinting platform was shown to be a customizable and efficient tool for assessing the effects of phosphate fertilizer on phosphate uptake, cell viability, growth, and root morphogenesis. This platform holds significant potential to contribute to the development of future precision fertilizer application technologies tailored to specific crop and soil conditions.

This work presents highly interesting and meaningful findings. The 3D bioprinting platform developed in this study is innovative and serves as a valuable initial screening tool.

However, for a more comprehensive evaluation of the efficacy of phosphate fertilizers, it is strongly recommended that plant-level experiments, utilizing the selected phosphate fertilizers, quantify their physiological effects by measuring dry weight (both above and below ground biomass) and phosphorus concentration within the plant body.

While changes in root morphology are important indicators of phosphorus deficiency response, they do not always directly correlate with overall plant growth or final yield. Therefore, measuring dry weight, as a direct indicator of biomass production, is essential. Furthermore, while FRET imaging is an excellent method for visualizing real-time phosphate dynamics at the cellular level, complementary measurements of the final plant phosphorus concentration are necessary to assess the absolute amount of absorbed phosphorus.

---

## [Reviewer Report]

Madison et al.’s work reports a new framework to assess phosphate fertilizer efficiency on cellular response and root tissue growth. They employed three-dimensional bioprinting in combination with protoplasts from BY-2 and Arabidopsis thaliana roots. The study is very well designed and executed. I find the findings important for future studies to investigate cellular phosphate uptake and response as basic science, as well as optimizing phosphorus fertilizer as applied agricultural science. I have some minor comments on this manuscript.

Line 176, Fig. da – typo

Line 183-184, The authors mentioned that their 3D bioprinting technique can recapitulate the cellular phosphate starvation response as observed in the intact root meristematic regions. I agree that cellular proliferation activity decreased in response to phosphate starvation, however, it remains unclear whether this is indeed what we see in the root meristematic regions or just looks like it. I recommend that the authors add some data to further discriminate this point or modify sentences for clarity.

Lines 194-196, The authors explain the results where the FRET biosensor activity decreases in response to the higher phosphate concentration. I am interested in whether the quantitative response of this FRET biosensor is consistent with the previous reports. This would be important to increase readability for the readers who has less specific knowledge on the biosensor.

---

## [Editor Report]

Dear authors,

We now have received the comments of the reviewers about your manuscript. Both reviewers found that the findings are important for future studies to investigate cellular phosphate uptake and response. Yet they raised few points of concern about the manuscript (please find the reviewers comments below).

Therefore and as a minimum requirement I would like to invite the authors to prepare and resubmit a moderately revised version of the manuscript in which they reviewers comments should be considered.

Thank you again for submitting your nice work to QPB.

We are looking forward to receiving your revised manuscript.

---

## [Reviewer Report]

Authors have addressed all the points I have suggested on their previous version. I have no more comments.

---

## [Editor Report]

Dear Dr. Rosangela Sozzani,

Your revised manuscript titled “Controlled delivery of phosphate to plants with optimized chemical and physical factors”has been reviewed, and I have now received the reviewers reports. Based on the reviewers feedback and on my own evaluation, I am happy to recommend the publication of your manuscript in QPB without further modification.

Thank you again for submitting your nice work to QPB.